# Association of low blood pressure and falls: An analysis of data from the Leiden 85-plus Study

David Röthlisberger[1], Katharina Tabea Jungo[1], Lukas Bütikofer[2], Rosalinde K. E. Poortvliet[3], Jacobijn Gussekloo[3,4], Sven Streit[1]*

1 Institute of Primary Health Care (BIHAM), University of Bern, Bern, Switzerland, 2 CTU Bern, University of Bern, Bern, Switzerland, 3 Department of Public Health and Primary Care, Leiden University Medical Center, Leiden, the Netherlands, 4 Department of Internal Medicine, Section Gerontology and Geriatrics, Leiden University Medical Center, Leiden, the Netherlands

☯ These authors contributed equally to this work.
* sven.streit@unibe.ch

**Data Availability Statement:** All relevant data are within the paper and its Supporting Information files.

## Abstract

### Background

Falls and consequent injuries are prevalent in older adults. In this group, half of injury-related hospitalizations are associated with falls and the rate of falls increases with age. The evidence on the role of blood pressure and the use of antihypertensive treatment on the risk of falls remains unclear in oldest-old adults (≥85 years).

### Objectives

To examine the association between systolic blood pressure (SBP) and incident falls with medical consequences in oldest-old adults and to analyse whether this association is modified by the use of antihypertensive treatments or the presence of cardiovascular disease.

### Methods

We analysed data from the Leiden 85-plus Study, a prospective, population-based cohort study with adults aged ≥85 years and a 5-year follow-up. Falls with medical consequences were reported by the treating physician of participants. We assessed the association between time-updated systolic blood pressure and the risk of falling over a follow-up period of five years using generalized linear mixed effects models with a binomial distribution and a logit link function. Subgroup analyses were performed to examine the role of antihypertensive treatment and the difference between participant with and without cardiovascular disease.

### Results

We analysed data from 544 oldest-old adults, 242 (44.4%) of which used antihypertensives. In 81 individuals (15%) ≥1 fall(s) were reported during the follow-up period. The odds for a fall decreased by a factor of 0.86 (95% CI 0.80 to 0.93) for each increase in blood pressure

**Funding:** The Leiden 85-plus Study was funded in part by an unrestricted grant from the Dutch Ministry of Health, Welfare and Sports. Prof. Streit's research is supported by grants (P2BEP3_165353) from the Swiss National Science Foundation (SNF) and the Gottfried and Julia Bangerter-Rhyner Foundation, Switzerland. These analyses did not receive any additional funding. The funders had no role in the study design, data collection and analysis, decision to publish, or preparation of the manuscript.

**Competing interests:** The authors declare that they have no competing interests.

by 10 mmHg. This effect was specific to blood pressure values above 130mmHg. We did not find any evidence that the effect would be modified by antihypertensive treatment, but that there was a tendency that it would be weaker in participants with cardiovascular disease (OR 0.81, 95% CI 0.72 to 0.90 per 10mmHg) compared to those without cardiovascular disease (OR 0.94, 95% CI 0.84 to 1.05 per 10mmHg).

## Conclusion

Our results point towards a possible benefit of higher blood pressure in the oldest-old with respect to falls independent of the use of antihypertensive treatments.

## Introduction

Hypertension is a common cardiovascular risk factor among adults aged 75 years and over [1]. For example, in the United States of America up to 75% of adults older than 75 years have been diagnosed with hypertension with numbers being slightly lower in Europe [2, 3]. The Joint National Committee (JNC8) defined hypertension as an elevated systolic blood pressure over 140 mmHg or elevated diastolic blood pressure over 90 mmHg [4]. Different observational studies showed an association between higher blood pressure and cardiovascular disease (CVD) and mortality [5, 6]. Therefore, low blood pressure is considered a protective factor for different diseases, such as CVD, renal insufficiency [7], retinopathy [8] and death, which is why hypertension should be treated [9, 10]. The treatment of hypertension in oldest-old patients is receiving greater attention, as the number of individuals within the oldest old age group within the general population is increasing rapidly as life expectancy increases. In this patient group, however, intensive blood pressure control and antihypertensive treatment can lead to overtreatment and treatment-related complications.

Different observational studies suggest that the positive relationship between high blood pressure and mortality is weakened in oldest-old adults [11–13]. For instance, a recent study showed that older frail adults with high systolic blood pressure (SBP) had a lower risk of all-cause mortality, suggesting that high SBP may be protective [14]. Further, the use of antihypertensive treatment increases adverse effects, such as dizziness, vertigo and falls [15]. Different studies showed that blood pressure lowered by antihypertensive treatment is considered a risk factor for first time falls and also recurrent falls in nursing home residents and in older adults living in assisted living in serviced apartments [16, 17]. One of the most commonly observed adverse effects from antihypertensive treatment and low blood pressure are falls and fall-related injuries [18]. Study results from the PARTAGE study showed a higher mortality risk for nursing home residents who took multiple blood pressure medications and achieved systolic blood pressure levels below 130 mmHg [19]. According to a report published by the World Health Organization [20], falls are a major public health problem in particular because they are associated with an increased mortality [21]. Therefore, the prevention of falls is of importance. However, the evidence on the association between falls and blood pressure in the oldest-old adults remains unclear.

In addition, the role of antihypertensive treatment on falls in oldest-old adults remains unclear. On the one hand, these medications may lead to changes in electrolyte levels or orthostatic hypotension, which could lead to falls in older adults [22]. On the other hand, there is evidence that antihypertensive treatment may improve cerebral blood flow and carotid distensibility and thus reduce the risk of falling [23, 24]. To contribute to filling the current research

gaps, the aim of this study was to study the association between systolic blood pressure (SBP) and incident falls in oldest-old adults and to analyze whether this association is modified by the use of antihypertensive treatments or a history of cardiovascular disease.

## Materials and methods

### Setting

For this study we used the data from the Leiden 85-plus Study, a prospective population-based cohort study of inhabitants of the city of Leiden in the Netherlands [25, 26]. As reported elsewhere, between September 1997 and September 1999 the study team enrolled 705 inhabitants of the city in the month of their 85th birthday [25, 26]. There were no exclusion criteria applied. From the target population of 705 inhabitants, 14 (2.0%) died before enrolment and a total of 566 participants gave their informed consent (or proxy consent was obtained) and were enrolled into the study [27]. In the present analyses, from the participants who provided informed consent to participate in the Leiden 85-plus Study, the data of 544 participants with any information on incident falls and blood pressure were analysed (96% of the cohort). As part of the Leiden 85-plus Study, participants were visited twice for face-to-face interviews and clinical examinations one month after their 85th birthday at the start of the study [27] and yearly thereafter with a follow-up of five years.

### Definitions of key variables and covariates

**Blood pressure.**   Blood pressure was measured yearly for all participants up to the age of 90 years. During these home visits, blood pressure was measured twice in a seated position with a mercury sphygmomanometer. Before measurement the participants were resting for at least five minutes and had not been exercising in the preceding 30 minutes. For the data analysis the mean of both measurements was calculated.

**Falls.**   The participants were followed-up for falls each year during a follow-up period of five years. At baseline and during the annual follow-up visits, the general practitioners or nursing home physicians were questioned about the medical history of the participants including their history of falls with medical consequences. For the present analyses, we used falls as a binary outcome (min. 1 falls vs. no falls).

**Antihypertensive treatment.**   Pharmacists provided a detailed listing of the medications used by participants. Antihypertensive treatment was defined using ATC (Anatomic Therapeutic Chemical) code groups. All medications included in the ATC code Group C02 (antihypertensives) were defined as antihypertensive treatment. Medication use was documented at baseline and during follow-up. For the present analyses, antihypertensive medication use was defined as a binary variable (use vs. no use) in each year of follow-up. We did not use any information on medication dosages.

**Other variables.**   Sociodemographic variables were recorded during the study visits (gender, education, income, living situation). For subgroup analyses the presence of CVD was documented (yes/no). CVD was defined as a diagnosis of either angina pectoris, myocardial infarction, heart failure, intermittent claudication, peripheral arterial surgery, transient ischemic attack (TIA) or stroke.

### Statistical analyses

We used descriptive statistics to report baseline characteristics of our sample stratified by antihypertensive treatment. The longitudinal data with one entry per year was modelled using generalized linear mixed effects models with a binomial distribution and a logit link function. We

used falls as a binary outcome (yes/no), the time-updated SBP as a fixed covariate, and added a random intercept for the participants to account for the correlation structure in the data. For the main analysis, we assumed a linear effect of SBP on the log odds for falls and the results are presented as multiplicative change in the odds for falls for an increase in SBP by 10 mmHg. As sensitivity analyses, to test whether the direction of the association was stable across the spectrum of mmHg values, we fitted piecewise linear models for SBP using linear splines with one or two knots and adjusted for the year of follow-up. For subgroup analyses, we added further covariates to the models including the time-updated intake of antihypertensives, history of CVD (at baseline) and death within the study, and their interaction with blood pressure. Further we stratified for frailty defined as lower than median hand grip strength or not able to perform a hand grip. Missing data was handled via the mixed effects models. All participants with non-missing information on blood pressure and falls on at least one time point were included in the main analysis. For participants who died or dropped out of the study, all observations up to the time point of death or drop-out were included. For all analyses we used Stata 16.0 (StataCorp, College Station, TX, USA).

## Ethics approval and consent to participate

The Medical Ethical Committee of the Leiden University Medical Center approved the original Leiden 85-plus Study. We confirm that all methods used in this study were carried out in accordance with the relevant guidelines and regulations. All participants provided informed consent before participating in the study, and all data were collected and analyzed according to applicable privacy and confidentiality laws.

## Results

In total, there were 3396 patient-years collected from the 544 included participants throughout the follow-up period at different timepoints. A total of 2249 patient-years from 544 participants had information on both falls and blood pressure and were used in our main analysis. Over the follow-up period of five years, we observed 340 patient-years in which at least one fall was reported, and we observed 1909 patient-years in which no falls were reported.

We observed an association between low SBP and a higher risk of falling (Fig 1). Overall, we found that the odds for a fall decreased by a factor of 0.86 (95% confidence interval (CI) 0.80 to 0.93) per 10 mmHg higher SBP. Using piecewise linear models, we found that the effect was only present in the group with SBP above 130 mmHg where the odds for falls decreased by 0.83 (95% CI 0.76 to 0.91) per 10 mmHg increase in SBP (S1 Fig in S1 File). Below 130mmHg there was no such association (OR: 1.64, 95% CI 0.93–2.91). Similar results were obtained after adjustment for the year of follow-up (S2 Fig in S1 File).

Of the included participants, 242 (44%) were prescribed antihypertensive treatment at baseline. Beside the fact that the subgroup without antihypertensive treatment had a smaller proportion of individuals with CVD (33.3% vs. 62.1%) the two subgroups were similar (Table 1). In 81 individuals (15%) one or more falls occurred in year before baseline. There was no difference in falls before baseline between the subgroup with or without antihypertensive treatment.

When comparing participants with and without antihypertensive treatment, we found in both groups a decreasing probability of falling with higher blood pressure (odds ratios of 0.81, 95% CI 0.72 to 0.91 without and 0.89, 95% CI 0.80 to 1.00 with antihypertensives, respectively) (Fig 2). We did not find evidence for an interaction between the use of antihypertensives and blood pressure (p for interaction = 0.248, S3 Fig in S1 File). When using spline models, we observed again that the effect was only present in the group with SBP above 130 mmHg. The results remained similar when we adjusted by the year of follow-up (S4 Fig in S1 File).

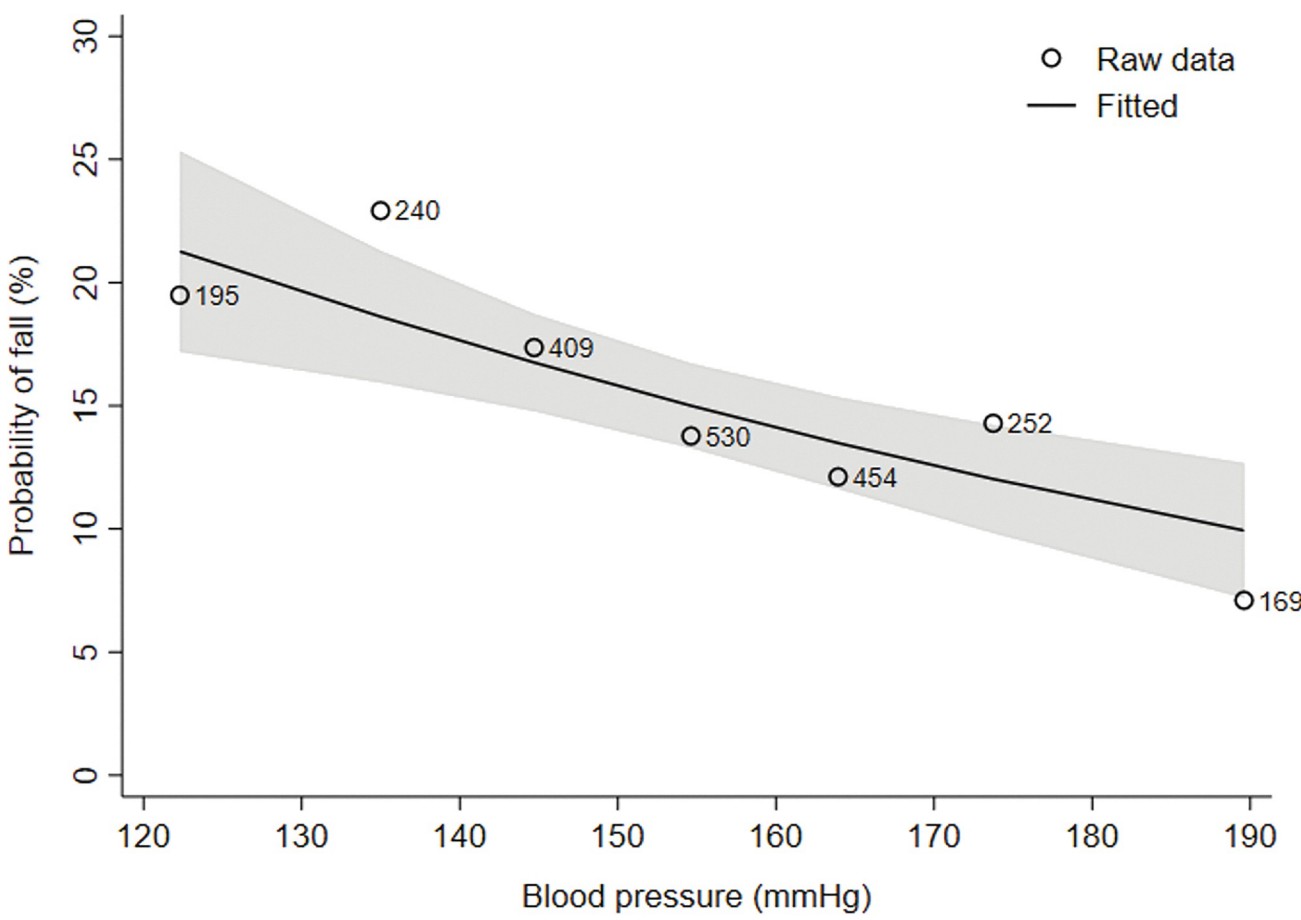

**Fig 1. Association between blood pressure and the probability of falls (n = 2249).** This figure shows the modelled probability to have at least one fall per year for different blood pressures (solid line) with 95% confidence band (shaded area). The points represent the raw data aggregated for blood pressures of <130, 130–140, 140–150, 150–160, 160–170 and >180 mmHg.

**Table 1. Baseline characteristics.**

| Domains | Overall (n = 544) | Antihypertensive treatment (n = 242) | No antihypertensive treatment (n = 302) | P-palue |
|---|---|---|---|---|
| Socio-demographic characteristics | | | | |
| Women, n (%) | 372 (65.7) | 170 (68.6) | 202 (63.5) | 0.21 |
| Low education[1], n (%) | 339 (64) | 158 (66.7) | 181 (61.2) | 0.27 |
| Low income[2], n (%) | 259 (49.2) | 144 (49) | 115 (49.4) | 0.93 |
| Institutionalized, n (%) | 78 (14.7) | 37 (15.6) | 41 (14) | 0.59 |
| Health characteristics | | | | |
| SBP in mmHg, mean (SD) | 155.4 (18.6) | 154.9 (16.7) | 155.8 (19.9) | 0.56 |
| CVD[3], n (%) | 260 (45.9) | 154 (62.1) | 106 (33.3) | <0.001 |
| At least 1 fall in year before baseline[4], n (%) | 81 (14.8) | 40 (16.5) | 41 (13.5) | 0.74 |

[1]defined as primary school only.

[2]defined as state pension only (about EUR 750 monthly)

[3]cardiovascular disease included angina pectoris, myocardial infarction, heart failure, intermittent claudication, peripheral arterial surgery, Transient ischemic attack (TIA) and stroke

[4] at least 1 fall in the year before baseline as reported by general practitioners of participants

Acronyms: CVD = cardiovascular disease, SBP = systolic blood pressure

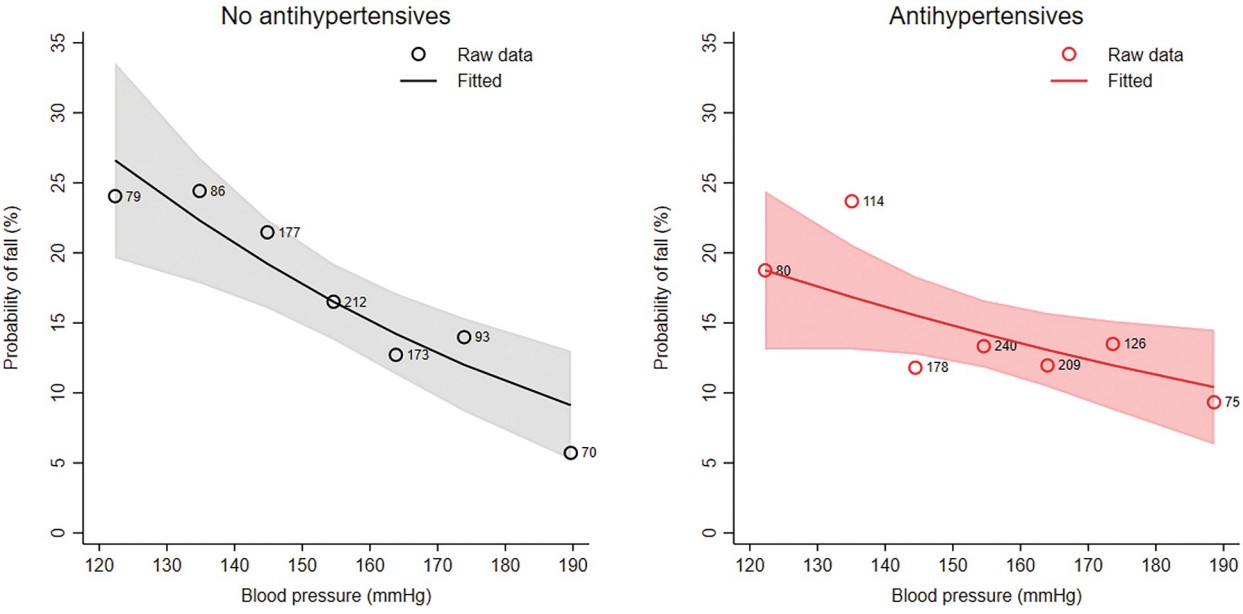

**Fig 2. Association between blood pressure and the probability of falls in participants with and without antihypertensives (without antihypertensives: n = 890 no antihypertensives; with antihypertensives: n = 1022).** The figures show the modelled probability to have at least one fall per year for different blood pressures (solid line) with 95% confidence band (shaded area). The points represent the raw data aggregated for blood pressures of <130, 130–140, 140–150, 150–160, 160–170 and >180 mmHg.

Fig 3 shows the association between blood pressure and the probability of falls in participants with and without CVD. In the subgroup of participants without a diagnosis of CVD the association between blood pressure and the probability to fall was more pronounced (odds ratio 0.81, 95% CI 0.72 to 0.90 per 10mmHg) than in participants with a diagnosis of CVD (odds ratio 0.94, 95% CI 0.84 to 1.05) (S8 Table in S1 File) (*p* for interaction 0.053, S9 Table in S1 File). The results remained similar when adjusting the models for the year of follow-up (S5 Fig in S1 File) and for frailty (S7 Fig in S1 File).

We did not find any evidence that the association of blood pressure and falls would depend on whether participants died during the study (S6 Fig in S1 File, p for interaction = 0.52). The odds for falls decreased by 0.86 (95% CI 0.77 to 0.94) and 0.90 (95% CI 0.79 to 1.01) per 10 mmHg for patients who did not and did die during the study, respectively.

## Discussion

This study aimed at investigating the association between systolic blood pressure and incident falls and to analyze whether this association is modified by the use of antihypertensive treatments or the presence of cardiovascular disease in older adults aged ≥85 years. We found that above the threshold of 130mmHg lower systolic blood pressure (SBP) values were associated with a higher risk of falling with the odds for falls decreasing by 0.83 (95% CI 0.76 to 0.91) per 10 mmHg. Below 130mmHg there was no such association (odds ratio: 1.35, 95% CI 0.81 to 2.26). In participants with and without antihypertensive treatment, the probability of falling decreased with higher blood pressure, particularly above SBP values of 130 mmHg. In the subgroup of participants without a history of CVD, the association between blood pressure and the probability to fall was more pronounced than in participants with a history of CVD. Overall, the results tend to point toward a possible benefit of higher systolic blood pressure in the age group of the oldest-old adults on the risk of falling.

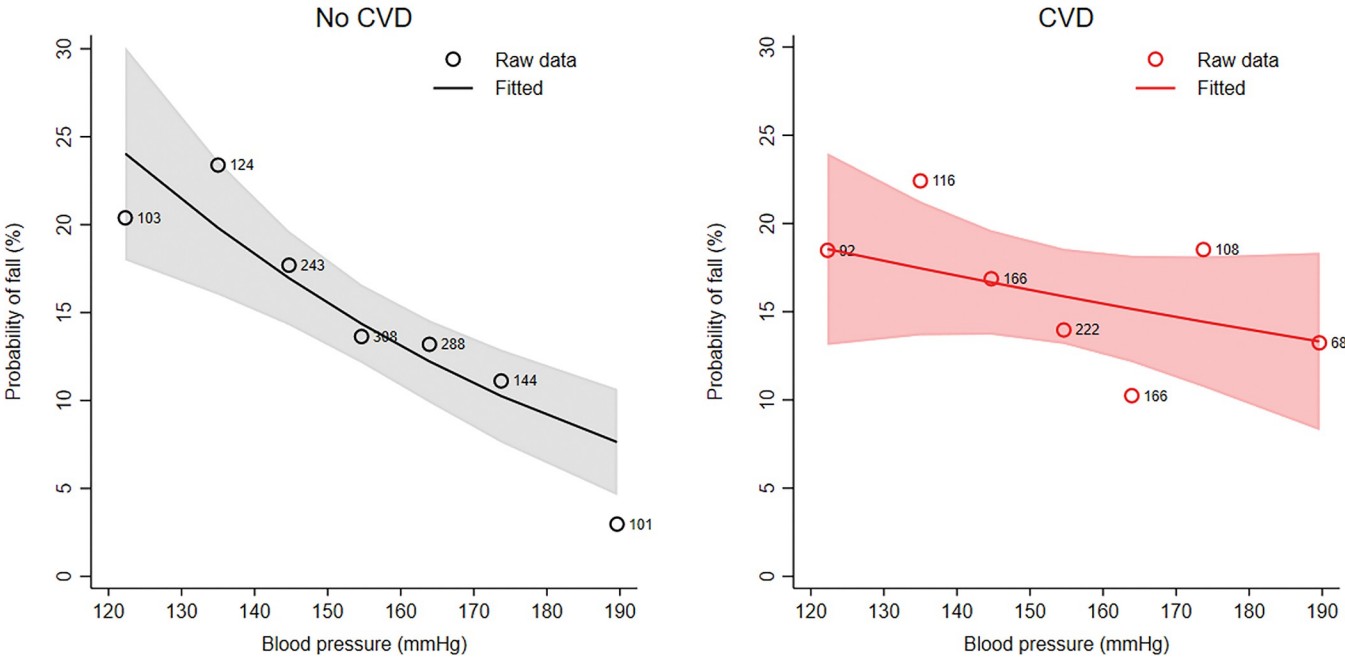

**Fig 3. Association between blood pressure and the probability of falls in participants with and without cardiovascular disease (CVD) (without CVD: n = 1300; with CVD: n = 938).** The figures show the modelled probability to have at least one fall per year for different blood pressures (solid line) with 95% confidence band (shaded area). The points represent the raw data aggregated for blood pressures of <130, 130–140, 140–150, 150–160, 160–170 and >180 mmHg.

In our study, we observed a lower risk of falling compared to other studies [28, 29]. One potential explanation for this discrepancy lies in the limitations of our study, which included only falls with medical consequences. This selective inclusion of falls with medical outcomes may have led to the observed lower risk, as it excludes less severe or non-medically relevant falls.

Our results are in line with previous studies on the association between systolic blood pressure and falls. For example, Klein [30] found, in a study with participants aged ≥60 years, an association with a decreased risk of falls in women with high blood pressure. At the same time, other observational studies point in the direction of low blood pressure being disadvantageous for older-old adults with regards to other patient outcomes. For instance, different studies showed that lower systolic blood pressure is associated with greater mortality in older adults aged 85 and over [12, 13].

There remains a lot of uncertainty around the ideal SBP targets in oldest-old adults. Mühl-bauer studied the literature relating to pharmacotherapy of hypertension in frail older patients and found conflicting results of randomized controlled trials and non-randomized studies, resulting in an absence of evidence-based recommendations. Nevertheless, the authors recommend that in patients over 80 years of age with a gait speed of less than 0.8 m/s target blood pressure values of 150 mmHg should be aimed for while in patients who are not more than mildly frail blood pressure values of 130-139mmHg should be aimed for [31].

Based on various studies, there is inconclusive evidence about the relationship between blood pressure and falls in different subgroups. For instance, a study from Sweden found that blood pressure and fall risk varied by functional status in patients over 60 years old [32]. In participants without functional impairment, high blood pressure was associated with a higher risk of falls. Low blood pressure was associated with an increased probability of falls in older

adults with functional impairment. Similarly, in a Chinese community of older adults with an SBP over 140 mmHg, Song found frailty to be a risk factor for falls [33]. Furthermore, the before mentioned study by Klein was able to show a gender-specific difference [30]. In persons 60 years and older the effects of hypertension on reducing falls risk were only observed in women. In contrast, low blood pressure levels (BP<120/80 mmHg) were associated with increased fall risk in men.

Different studies investigated the association between antihypertensive treatment and falls. However, most previous research was done in 'younger' populations of older adults compared to the study population in our analyses (≥85 years). Similar to our findings, Zachary did not find an overall association between the use of antihypertensive treatment and recurrent falls in participants aged 70–79 years [34]. A meta-analysis from 2013 did not find an association between the use of any antihypertensive treatment and an increased risk of fall injuries in the oldest old either [35]. However, a case-crossover study with over 90'000 participants, found a short-term, but no long-term increased risk of fall injuries after antihypertensive treatment initiation in participants ≥65 years old [22]. Similarly, Tinetti et al. [36] found a significant association between the use of antihypertensive treatment and an increased risk of serious fall injuries. These findings seem to differ from ours. While we investigated the risk of falls with medical consequences that were reported by participants and their general practitioners, they defined serious fall injuries based on emergency department visits recorded in claims data. This difference, together with the possibility that participants in our study may have used antihypertensive treatment for a longer period and those with falls may have already stopped it, may be one reason for the different results.

## Strengths and limitations

With a high inclusion rate and almost complete follow-up, this study benefits from being based on a population-based sample of oldest old adults. The generalisability of this study is further enhanced by including data from participants living in nursing homes and not only community-dwelling older adults. Further, it is strengthened by the fact that the study design allowed to establish a temporal relationship between time-updated SBP values measured and outcome assessments over the follow-up of five years.

We would like to highlight some limitations. First, this is an observational study with data collected from 1997 onwards and therefore has limitations due to its study design. For example, the relationship between exposure and outcome may be biased or confounded by unmeasurable or uncontrollable factors (such as orthostatic hypotension or psychotropic medication for example). However, our sensitivity analyses show similar results. Secondly, we assumed data missing at random in the mixed-effects models. That might not be the case, especially for missingness due to death. Survivorship bias therefore cannot be excluded. However, we did not find any evidence that death during the follow-up period would modify the effect of blood pressure on falls. The third limitation was the focus on falls with medical consequences that were reported by general practitioners, which may have led to an underestimation of the overall number of falls. Since falls were assessed yearly, there was a maximum time delay of one year for recording a fall. Finally, a lack of continuous drug monitoring and absence of dosing information are other limitations of this study.

## Conclusion

The findings of this study demonstrate that above the threshold of 130 mmHg lower systolic blood pressure values are associated with a higher risk of falls with medical consequences in adults aged 85 years and older independent of the use of antihypertensive treatment. Again,

the effect was present in the group with SBP above 130 mmHg but not below. Our results thus point towards a possible benefit of higher blood pressure in the oldest-old with respect to falls.

## Supporting information

**S1 File.**
(DOCX)

## Author Contributions

**Conceptualization:** Rosalinde K. E. Poortvliet, Jacobijn Gussekloo, Sven Streit.

**Data curation:** David Röthlisberger, Katharina Tabea Jungo, Lukas Bütikofer, Rosalinde K. E. Poortvliet, Jacobijn Gussekloo, Sven Streit.

**Formal analysis:** David Röthlisberger, Katharina Tabea Jungo, Lukas Bütikofer, Rosalinde K. E. Poortvliet, Jacobijn Gussekloo, Sven Streit.

**Investigation:** David Röthlisberger, Katharina Tabea Jungo, Rosalinde K. E. Poortvliet, Jacobijn Gussekloo, Sven Streit.

**Methodology:** David Röthlisberger, Katharina Tabea Jungo, Lukas Bütikofer, Rosalinde K. E. Poortvliet, Jacobijn Gussekloo, Sven Streit.

**Project administration:** Rosalinde K. E. Poortvliet, Jacobijn Gussekloo, Sven Streit.

**Resources:** Katharina Tabea Jungo, Rosalinde K. E. Poortvliet, Sven Streit.

**Software:** Lukas Bütikofer.

**Supervision:** Katharina Tabea Jungo, Rosalinde K. E. Poortvliet, Jacobijn Gussekloo, Sven Streit.

**Validation:** David Röthlisberger, Katharina Tabea Jungo, Jacobijn Gussekloo, Sven Streit.

**Visualization:** David Röthlisberger, Katharina Tabea Jungo, Sven Streit.

**Writing – original draft:** David Röthlisberger, Katharina Tabea Jungo, Lukas Bütikofer, Rosalinde K. E. Poortvliet, Sven Streit.

**Writing – review & editing:** David Röthlisberger, Katharina Tabea Jungo, Lukas Bütikofer, Rosalinde K. E. Poortvliet, Jacobijn Gussekloo, Sven Streit.

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
