## [Decision Letter · Decision Letter 0]

2 Nov 2023

PONE-D-23-23975Association of low blood pressure and falls: An analysis of data from the Leiden 85-plus StudyPLOS ONE

Dear Dr. Streit,

Thank you for submitting your manuscript to PLOS ONE. After careful consideration, we feel that it has merit but does not fully meet PLOS ONE’s publication criteria as it currently stands. Therefore, we invite you to submit a revised version of the manuscript that addresses the points raised during the review process.

The manuscript is necessary to be major-revisioned according to the Reviewer's comments.

We look forward to receiving your revised manuscript.

Kind regards,

Masaki Mogi

Academic Editor

PLOS ONE

“The Leiden 85-plus Study was funded in part by an unrestricted grant from the Dutch Ministry of Health, Welfare and Sports. Prof. Streit’s research is supported by grants (P2BEP3_165353) from the Swiss National Science Foundation (SNF) and the Gottfried and Julia Bangerter-Rhyner Foundation, Switzerland. These analyses did not receive any additional funding.”

4. Please include your tables as part of your main manuscript and remove the individual files. Please note that supplementary tables (should remain/ be uploaded) as separate "supporting information" files

Reviewers' comments:

Reviewer's Responses to Questions

**Comments to the Author**

1. Is the manuscript technically sound, and do the data support the conclusions?

Reviewer #1: Yes

2. Has the statistical analysis been performed appropriately and rigorously? 

Reviewer #1: No

3. Have the authors made all data underlying the findings in their manuscript fully available?

Reviewer #1: Yes

4. Is the manuscript presented in an intelligible fashion and written in standard English?

Reviewer #1: Yes

5. Review Comments to the Author

Reviewer #1: With interest I have read the manuscript.

In my opinion, the manuscript may benefit from an elaboration of the authors regarding the limitations of their study design by comparing their findings with other published data. For example, only 15% of the study sample experienced at least one fall. Data show that 30-40% of adults older than 65 years fall yearly (https://www.acpjournals.org/doi/full/10.7326/0003-4819-157-3-201208070-00462). In their study in the oldest-old, only 15% had reported to have fallen in the previous year. This is a large difference that should be emphasized more. See also: https://www.tandfonline.com/doi/full/10.2147/CIA.S57580.

In addition, the authors have only corrected for a small list of confounders without justification of the selection. Where is this based on? Falls are multifactorial in nature, and it seems like some major fall-risk factors such as orthostatic hypotension, frailty/dependence, and non-antihypertensive concomitant medications (e.g., psychotropic medications: https://link.springer.com/article/10.1007/s40266-014-0225-x) are not considered?

Also, reading the manuscript, reflection on potential differences between antihypertensive (sub)classes in associations with fall risk (e.g., https://www.sciencedirect.com/science/article/pii/S1525861017306989) are missing. Was it possible to do subgroup analyses based on antihypertensive (sub)class?

Is it possible to calculate time to first fall and/or number of falls instead of any fall during follow-up? This may be more informative.

Lastly, I would like to advice the authors to thoroughly review the main text of the manuscript for typo's (e.g., person instead of persons in lines 270-271), odd phrasing (example lines 163-164 "in older patients with functional status greater than 60 years old"), omissions (line 80: "values" missing before blood pressure?) and duplications (example: "however" in lines 76-77).

6. PLOS authors have the option to publish the peer review history of their article (what does this mean?). If published, this will include your full peer review and any attached files.

Reviewer #1: No

---

## [Author Response · Author response to Decision Letter 0]

24 Nov 2023

Point-by-point Response

Response: We have reviewed the relevant guidelines and adjusted the documents as per your requirements.

“The Leiden 85-plus Study was funded in part by an unrestricted grant from the Dutch Ministry of Health, Welfare and Sports. Prof. Streit’s research is supported by grants (P2BEP3_165353) from the Swiss National Science Foundation (SNF) and the Gottfried and Julia Bangerter-Rhyner Foundation, Switzerland. These analyses did not receive any additional funding.” Please state what role the funders took in the study. If the funders had no role, please state: "The funders had no role in study design, data collection and analysis, decision to publish, or preparation of the manuscript." If this statement is not correct you must amend it as needed. Please include this amended Role of Funder statement in your cover letter; we will change the online submission form on your behalf.

Response: We have adjusted the "financial disclosure" section in accordance with your instructions. Thank you for your guidance. It now reads: 

The Leiden 85-plus Study was funded in part by an unrestricted grant from the Dutch Ministry of Health, Welfare and Sports. Prof. Streit’s research is supported by grants (P2BEP3_165353) from the Swiss National Science Foundation (SNF) and the Gottfried and Julia Bangerter-Rhyner Foundation, Switzerland. These analyses did not receive any additional funding. The funders had no role in the study design, data collection and analysis, decision to publish, or preparation of the manuscript.

Response: Thank you for your feedback. We have revised the manuscript as per your request, ensuring that the ethics statement now appears exclusively in the Methods section (p. 8).

4. Please include your tables as part of your main manuscript and remove the individual files. Please note that supplementary tables (should remain/ be uploaded) as separate "supporting information" files. 

Response: We've included the tables in the main manuscript and removed the individual files, as requested. Supplementary tables are now in separate "supporting information" files.

Response: We have included captions for the Supporting Information files at the end of our manuscript and updated in-text citations to match accordingly.

 

Reviewers' comments:

1. Is the manuscript technically sound, and do the data support the conclusions?

Reviewer #1: Yes

Response: Thank you for your review and assessment of our manuscript. We appreciate your feedback and are pleased to hear that you found our research technically sound and concluded appropriately based on the presented data.

2. Has the statistical analysis been performed appropriately and rigorously?

Reviewer #1: No

Response: We have taken note of the reviewer's comment, and we reviewed and made the necessary adjustments to ensure that the statistical analysis is performed appropriately and rigorously. The detailed description of these modifications can be found in the second part of the cover letter. 

3. Have the authors made all data underlying the findings in their manuscript fully available?

Reviewer #1: Yes

Response: Thank you for your evaluation. 

4. Is the manuscript presented in an intelligible fashion and written in standard English?

Reviewer #1: Yes

Response: Thank you for your assessment. 

5. Review Comments to the Author

Comment 1:

In my opinion, the manuscript may benefit from an elaboration of the authors regarding the limitations of their study design by comparing their findings with other published data. For example, only 15% of the study sample experienced at least one fall. Data show that 30-40% of adults older than 65 years fall yearly (https://www.acpjournals.org/doi/full/10.7326/0003-4819-157-3-201208070-00462). In their study in the oldest-old, only 15% had reported to have fallen in the previous year. This is a large difference that should be emphasized more. See also: https://www.tandfonline.com/doi/full/10.2147/CIA.S57580.

Response: Thank you for this comment. We appreciate your suggestion to elaborate on the limitations of our study design and to compare our findings with other published data. We have taken your feedback into account, and in our revised manuscript, we have included a discussion of the observed differences in fall rates between our study and existing literature. In our conclusion and key points, we specifically tailored the text to highlight falls with medical consequences (as measured in our study), acknowledging and addressing this distinction to provide a more nuanced interpretation of our findings, which may contribute to the observed lower risk of falling compared to other studies.

Manuscript Change: The text on p. 13, Line 290-294 now reads: 

In our study, we observed a lower risk of falling compared to other studies. One potential explanation for this discrepancy lies in the limitations of our study, which included only falls with medical consequences. This selective inclusion of falls with medical outcomes may have led to the observed lower risk, as it excludes less severe or non-medically relevant falls.

The text on p. 2 Line 64-66 now reads:

We found that above the threshold of 130mmHg higher systolic blood pressure values were associated with a lower risk of falls with medical consequences in oldest-old adults, independent of the use of antihypertensive medications.

The text on p. 16 Line 381-383 now reads:

The findings of this study demonstrate that above the threshold of 130 mmHg lower systolic blood pressure values are associated with a higher risk of falls with medical consequences in adults aged 85 years and older independent of the use of antihypertensive treatment.

Comment 2:

In addition, the authors have only corrected for a small list of confounders without justification of the selection. Where is this based on? Falls are multifactorial in nature, and it seems like some major fall-risk factors such as orthostatic hypotension, frailty/dependence, and non-antihypertensive concomitant medications (e.g., psychotropic medications: https://link.springer.com/article/10.1007/s40266-014-0225-x) are not considered?

Response: 

We agree with the reviewer’s comment that several factors can lead to a fall. However it is not possible to correct for each and every one. We therefore needed to select confounders that are a) relevant and b) available in the dataset. As described on page 7, we chose beside socio-economic characteristics factors like the presence of a cardiovascular disease (CVD) and allowed our models to analyse time-updated information of antihypertensive medication. Because we do not have time-updated data on orthostatic hypotension or psychotropic medication we updated our limitations section. However, we can include a stratification for frailty where the results did not differ for patients with or without frailty (frailty defined as lower than median hand grip strength or not able to perform a hand grip). 

This leads to the following changes:

Manuscript Change: The Text on p.7 Line 183-184 now reads:

Further we stratified for frailty defined as lower than median hand grip strength or not able to perform a hand grip.

The Text on p.11 Line 255-257 now reads:

The results remained similar when adjusting the models for the year of follow-up (S5 Fig) and for frailty (S7 Fig).

The Text on p. 15 Line 353-356 now reads:

For example, the relationship between exposure and outcome may be biased or confounded by unmeasurable or uncontrollable factors (such as orthostatic hypotension or psychotropic medication for example). 

Comment 3:

Also, reading the manuscript, reflection on potential differences between antihypertensive (sub)classes in associations with fall risk (e.g., https://www.sciencedirect.com/science/article/pii/S1525861017306989) are missing. Was it possible to do subgroup analyses based on antihypertensive (sub)class?

Response: We appreciate your comment and the suggestion for subgroup analyses based on antihypertensive subclasses. 

It's important to note that our study was not specifically designed to delve into the nuances of different types of blood pressure-lowering medications. One contributing factor to this decision is the common practice of prescribing these medications in combination, making it challenging to attribute effects to individual subclasses accurately. Also, one would need to incorporate dosage, dosage over time and other factors such as kidney function. Our primary focus was on understanding the broader relationship between blood pressure levels and falls within the limitations of the available dataset however in a highly representative cohort with time-updated information on blood pressure and a long follow-up. 

Furthermore, it's worth noting that creating subgroups based on antihypertensive subclasses would likely result in small sample sizes, limiting the statistical power and reliability of any conclusions drawn in this context.

Thank you for your understanding and consideration. 

Comment 4:

Is it possible to calculate time to first fall and/or number of falls instead of any fall during follow-up? This may be more informative.

Response: The information on falls was collected annually without dates. Therefore we chose to annually analyze data on falls. 

Comment 5

Lastly, We would like to advice the authors to thoroughly review the main text of the manuscript for typo's (e.g., person instead of persons in lines 270-271), odd phrasing (example lines 163-164 "in older patients with functional status greater than 60 years old"), omissions (line 80: "values" missing before blood pressure?) and duplications (example: "however" in lines 76-77).

Response: Thank you for your valuable feedback. We have carefully reviewed the manuscript and addressed the issues you pointed out, including typos, odd phrasing, omissions, and duplications. We have made the necessary revisions to enhance the clarity and quality of the text.

Manuscript Change: 

The text on p.14 , Line 466-468 now reads: 

In persons 60 years and older the effects of hypertension on reducing falls risk were only observed in women.

The text on p.14 , Line 320-322 now reads: 

For instance, a study from Sweden found that blood pressure and fall risk varied by functional status in patients over 60 years old.

The text on p.4 , Line 88-89 now reads: 

Different observational studies suggest that the positive relationship between high blood pressure and mortality is weakened in oldest-old adults.

The text on p.4 , Line 85-87 now reads: 

In this patient group, however, intensive blood pressure control and antihypertensive treatment can lead to overtreatment and treatment-related complications.

---

## [Decision Letter · Decision Letter 1]

4 Dec 2023

Association of low blood pressure and falls: An analysis of data from the Leiden 85-plus Study

PONE-D-23-23975R1

Dear Dr. Streit,

We’re pleased to inform you that your manuscript has been judged scientifically suitable for publication and will be formally accepted for publication once it meets all outstanding technical requirements.

Kind regards,

Masaki Mogi

Academic Editor

PLOS ONE

Additional Editor Comments (optional):

Reviewers' comments:

Reviewer's Responses to Questions

**Comments to the Author**

1. If the authors have adequately addressed your comments raised in a previous round of review and you feel that this manuscript is now acceptable for publication, you may indicate that here to bypass the “Comments to the Author” section, enter your conflict of interest statement in the “Confidential to Editor” section, and submit your "Accept" recommendation.

Reviewer #1: All comments have been addressed

2. Is the manuscript technically sound, and do the data support the conclusions?

Reviewer #1: Yes

3. Has the statistical analysis been performed appropriately and rigorously? 

Reviewer #1: Yes

4. Have the authors made all data underlying the findings in their manuscript fully available?

Reviewer #1: Yes

5. Is the manuscript presented in an intelligible fashion and written in standard English?

Reviewer #1: Yes

6. Review Comments to the Author

Reviewer #1: (No Response)

7. PLOS authors have the option to publish the peer review history of their article (what does this mean?). If published, this will include your full peer review and any attached files.

Reviewer #1: **Yes: **Eveline van Poelgeest

---

## [Editor Report · Acceptance letter]

11 Dec 2023

PONE-D-23-23975R1 

Association of low blood pressure and falls: An analysis of data from the Leiden 85-plus Study 

Dear Dr. Streit:

I'm pleased to inform you that your manuscript has been deemed suitable for publication in PLOS ONE. Congratulations! Your manuscript is now with our production department. 

Kind regards, 

on behalf of

Dr. Masaki Mogi 

Academic Editor

PLOS ONE